

# Semantic plausibility preferentially affects the semantic preview benefit in Chinese reading: evidence from an eye-movement study

Lijuan Zhang[1], Zhiwei Liu[2], Sainan Zhao[3] and Jingxin Wang[1]

[1] Faculty of Psychology, Tianjin Normal University, Tianjin, China
[2] School of Education and Psychology, Sichuan University of Science and Engineering, Zigong, Sichuan, China
[3] School of Education, Linyi University, Linyi, Shandong, China

## ABSTRACT

**Background:** Numerous studies have confirmed that skilled readers can benefit from a semantically related preview word (*i.e.*, semantic preview benefit, SPB), suggesting that readers can extract semantic information from the parafovea to achieve efficient reading. It is still under debate whether the occurrence of this benefit is because of the semantic association between the preview and target words or because of the contextual fit of the preview word in the sentence context.
**Methods:** Two independent factors, preview plausibility (preview plausible/implausible) and semantic relatedness (semantically related/unrelated), were manipulated, and we further strictly controlled for syntactic plausibility in the present study.
**Results:** The results showed that the first-pass reading times of the target words were significantly shorter in the plausible preview condition than in the implausible preview condition. However, the main effect of semantic relatedness was found only in the gaze duration measure.
**Discussion:** The pattern of results revealed that semantic plausibility affects the semantic preview benefit preferentially in Chinese reading, supporting the contextual fit account. Our findings have implications for a better understanding of parafoveal processing and provide empirical support for the eye-movement control model.

# INTRODUCTION

A large number of studies have confirmed that skilled readers can preview the upcoming word that appears to the right of the fixation point (the parafoveal vision), which facilitates the processing of the word when it subsequently becomes fixated on (*Rayner, 1998*, *2009*). The facilitation resulting from the preprocessing of the parafoveal word is called the preview benefit (PB), which can be observed in fixation duration measures (*Rayner, 1998*, *2009*). Several studies have shown that readers can obtain an orthographic and phonological preview benefit, suggesting that readers can extract sublexical information from parafoveal vision during sentence reading (*Schotter, Angele & Rayner, 2012*). Some

Corresponding author
Jingxin Wang, wjxpsy@126.com

studies have found that readers can also extract lexical information from the parafovea and obtain the semantic preview benefit (SPB; *Hohenstein, Laubrock & Kliegl, 2010*; *Schotter, 2013*; *Yan et al., 2009*, *2012*). The controversy over the underlying mechanisms of SPB has recently raged unabatedly (*Li, Sun & Wang, 2022*; *Schotter & Jia, 2016*; *Veldre & Andrews, 2016*). It mainly concerns whether SPB occurs because of semantic associations between the preview and target words or reflects context fit between the preview word and the sentence. Based on previous studies, the present study aims to explore the underlying mechanisms of SPB by further controlling syntactic plausibility.

Eye-tracking technology and the gaze-contingent boundary paradigm (*Rayner, 1975*) were used to investigate the SPB. In a standard boundary paradigm, an invisible boundary is set between the foveal pretarget word and the parafoveal target word. A preview stimulus (a preview word that is identical, semantically related, or unrelated to the target word) is presented in the target position and changes to the target word when the readers make a saccade across the invisible boundary. Generally, SPB is observed when the first-pass reading times in the semantically related preview condition are significantly shorter than those in the semantically unrelated preview condition (*Schotter, Angele & Rayner, 2012*), suggesting that semantic information can be obtained from the parafovea and that previewing a semantically related word can facilitate the processing of the target word than previewing a semantically unrelated word. Using the boundary paradigm, the SPB has been found in alphabetic and logographic writing systems such as German (*Hohenstein, Laubrock & Kliegl, 2010*), Finnish (*White, Bertram & Hyönä, 2008*), English (*Schotter, 2013*), and Chinese (*Yan et al., 2009*, *2012*; *Yang et al., 2012*).

A much-debated question is why SPB is observed in reading. There are two explanations for the SPB reported in reading studies. One explanation is that SPB may arise from the semantic associations between the preview and target words. The processing of the semantically related word would directly activate the target word, which is similar to the underlying mechanism of semantic priming effect (*Neely, 1991*). Early researchers proposed that a priming word (a semantically related word) would activate relevant semantic features that are shared with the target word through semantic spreading and facilitate the processing of target words (*Collins & Loftus, 1975*; *Hutchinson, 2003*). Thus, processing a semantically related preview preactivates semantic information relative to the target word and then facilitates the subsequent processing of the target word compared to a semantically unrelated word (*Schotter, 2013*). For example, several studies have found that a semantically related or highly similar preview word (synonym) reduced the fixation durations on the target word relative to a semantically unrelated preview word, exhibiting the SPB effect (*Hohenstein, Laubrock & Kliegl, 2010*; *Schotter, 2013*; *Yan et al., 2009*, *2012*).

Another alternative theory, however, argues that the SPB may reflect the extent to which the preview word contextually fits with the sentence, rather than the semantic association between the preview and target words (*Yang et al., 2012*; *Schotter & Jia, 2016*; *Veldre & Andrews, 2016*). On this account, previews that contextually fit with the preceding sentence context (the plausible preview) facilitate the processing of target words, whereas an implausible word leads to a cost. For example, *Yang et al. (2012)* previously reported a significant plausibility preview benefit in the early stage, in which the fixation durations of

the target word were shorter in the plausible preview condition than in the implausible preview condition. In addition, *Yang et al. (2012)* found that readers can benefit from semantically related previews when the preview word is plausible (experiment 2) but not when the preview word is implausible (experiment 1). In a recent study, *Li, Sun & Wang (2022)* used high-constraint sentences also demonstrated that the semantic relatedness preview benefit was observed only when the preview word was plausible, suggesting that preview plausibility plays an important role in the SPB in Chinese reading. In English reading, researchers have also discovered that the preview plausibility is of great significance to parafoveal semantic processing (*Antúnez et al., 2022*; *Schotter & Jia, 2016*; *Veldre & Andrews, 2016*). For example, *Veldre & Andrews (2016)* investigated the underlying mechanisms of SPB by orthogonally manipulating preview plausibility and semantic relatedness. Their results showed that there was a significant plausibility preview benefit, regardless of the semantic relationship between the preview and target word; importantly, no additional benefit was observed in the semantically related preview condition than in the unrelated preview condition when the previews were plausible. In conclusion, the preview plausibility and semantic relatedness of the preview and target words affect the SPB in reading. To date, the underlying mechanism behind SPB occurrence in reading is still unclear.

Compared to alphabetic writing systems, Chinese as a logographic writing system has distinct characteristics (*Li et al., 2015*; *Li, Sun & Wang, 2022*) that provide advantages for investigating this issue. For example, Chinese is formed by strings of square characters that each take up the same amount of space. There is no space between Chinese characters, which may potentially improve the possibility for readers to extract semantic information from the parafovea (*Yang et al., 2012*). On the other hand, previous studies on SPB in Chinese reading did not take into account the preview plausibility (*Yan et al., 2009*, *2012*; *Pan, Laubrock & Yan, 2016*); or, despite considering the plausibility of the preview word (*Li, Sun & Wang, 2022*; *Yang et al., 2012*), did not manipulate the preview plausibility and semantic relatedness orthogonally to explore their combined effect on SPB. This manipulation may cause the SPB results to be obscure and unstable. Thus, further verification and investigation are needed to determine the stability of the SPB results in Chinese reading. The present study was designed to investigate these knowledge gaps. Based on previous research (*Li, Sun & Wang, 2022*; *Yang et al., 2012*), we investigated whether the SPB was derived from the semantic associations or rather reflected the plausibility of the preview word and sentence context by orthogonally manipulating the preview plausibility and semantic relatedness.

In addition, previous research in the field paid little attention to syntactic plausibility, which may have contributed to the inconsistent findings and the debate over the SPB effect between the two accounts mentioned above. Specifically, previous studies on the SPB did not rule out the effect of syntactic plausibility on the preview benefit. While related/plausible previews are always syntactically plausible, unrelated/implausible previews may not be (*Schotter, 2013*; *Yan et al., 2009*, *2012*; *Yang et al., 2012*). The difference in the syntactic plausibility of preview conditions may make the observed preview benefit ambiguous. For instance, in the study of *Veldre & Andrews (2016)*, approximately 55% of

implausible previews violate syntactic structure rules (*e.g.*, the preview should be a noun, but it was a verb). For example, in the sentence "*Toby kept his money in a large barn/woke because he lived on a farm*", the implausible and unrelated preview "*woke*" was not only semantically implausible but also constituted a syntactical violation. Therefore, the SPB effect found in previous studies may also reflect readers' sensitivity to syntactic plausibility. That is, the SPB reported in previous research might result from syntactic plausibility in semantically related or contextually plausible preview conditions, but not arise entirely from semantic spreading between the preview and target words nor the fitted context of plausible previews.

Recently, the effects of syntactic information on parafoveal lexical processing in reading have been demonstrated in eye-movement studies (*Veldre & Andrews, 2018*). *Veldre & Andrews (2018)* explored the effects of syntactic and semantic plausibility on parafoveal processing in English reading. They found that syntactically violated and semantically implausible previews interfered with the first-pass reading of the target word to a greater degree than syntactically plausible but semantically implausible previews (*she eventually found a spare stool/uncle/begin behind the crowded bar.*). Moreover, the first-pass reading times of the target words could be significantly shortened when their previews were syntactic and semantically plausible compared to when their previews were syntactic violations but semantically plausible (*her plane will probably refuel/depart/landed later than expected this afternoon*). The pattern of results showed that readers can extract syntactic information from the parafovea at a relatively early stage and benefit from syntactic acceptability in parafoveal processing.

In Chinese reading, several studies have shown that syntactic cues play a minor role in reading than that semantics and context information (*Li, 1996*; *Li, Bates & MacWhinney, 1993*). However, other studies have suggested that a violation of syntactic information interferes severely with reading, indicating that syntactic information can be processed very early in reading (*Yang et al., 2009*; *Yu & Zhang, 2008*). For example, *Yang et al. (2009)* used eye-tracking technology to investigate the temporal process of syntactic and semantic processing in sentence reading by setting three conditions for the target words: semantically and syntactically anomalous, semantically anomalous, and congruent. Their results showed that the first-pass reading times for the target words were significantly shorter in the congruent condition than in the other two conditions, but semantic and syntactic violations caused more serious disruptions than semantic violations did, suggesting that syntax plays an important role and that syntactic violations could cause disruptions in Chinese reading (*Yang et al., 2009*). Therefore, syntactic processing can occur at an early stage during reading, which could exert an effect on the preview benefit. For example, in previous studies on SPB, the semantically implausible or unrelated preview words may also violate syntactic rules (syntactically implausible) (*Li, Sun & Wang, 2022*; *Yang et al., 2012*; *Zhu, Zhuang & Ma, 2021*), which might confuse the SPB results. Consequently, syntactic plausibility needs to be controlled when investigating the SPB effect. Based on previous studies (*Li, Sun & Wang, 2022*; *Yang et al., 2012*; *Zhu, Zhuang & Ma, 2021*), the present study restricted all preview words to be syntactically plausible to

avoid the interference of syntactically implausible previews with parafoveal semantic processing.

The present study aimed to investigate the underlying mechanisms of the SPB by using eye-tracking technology and the boundary paradigm. Two independent factors, preview plausibility (plausible/implausible) and semantic relatedness (semantically related/ unrelated) were manipulated, and syntactic plausibility was strictly controlled in the present study. Our study does not aim to systematically investigate the effect of parafoveal syntactic information on SPB, but rather aims to further explore the results pattern of SPB by excluding the interference of syntactically implausible information based on previous studies (*Li, Sun & Wang, 2022*; *Yang et al., 2012*; *Zhu, Zhuang & Ma, 2021*). Therefore, the preview words in the manipulated conditions were all syntactically plausible to ensure that the syntactic processing of the preview words could not confuse the SPB effect in the present study. If the SPB effect reflected semantic associations between the preview and target words, then the semantic relatedness effect should be observed regardless of whether the previews are plausible. Moreover, if the SPB effect reflects the contextual fit of the preview with the preceding sentence, then the plausibility preview effect should occur regardless of whether the preview word is semantically related to the target word. We believe that the underlying mechanisms behind SPB will be further understood in the present study.

## METHOD

### Ethical approval

This study was approved by the Research Ethics Committee of the Faculty of Psychology, Tianjin Normal University (Approval Number, 2022092104), and conducted in accordance with the principles of the Declaration of Helsinki. All participants volunteered to participate in the experiment and signed informed consent.

### Participants

G * power 3.1 software was used to calculate the number of subjects in the two-factor within-subject experimental design. When the effect size reached $f = 0.25$ and the significance level reached $\alpha = 0.05$, a total of 54 subjects were needed to achieve 95% statistical test power. Compared with the 20 sentences under each condition in previous studies (*Li, Sun & Wang, 2022*; *Zhu, Zhuang & Ma, 2021*), there are only 16 sentences used under each condition in our study. To retain as many trials as possible for analysis, we recruited 80 students ($M = 20.15$ years, $SD = 1.60$; 15 males) from Tianjin Normal University to participate in the eye-tracking experiment. All participants were native Chinese speakers with normal or corrected-to-normal vision. Participants were paid 30 RMB for their participation in the experiment.

### Materials and design

Note that most Chinese words (72%) are two-character words (*Wei, Li & Pollatsek, 2013*); therefore, 80 representative two-character words were selected as targets. To minimize the repetition of sentence structures (*Veldre & Andrews, 2018*), the target words come from

**Table 1 An example of sentence materials used in experiment.**

| Preview conditions | Sentence example |
| --- | --- |
| Identical | 粗心的佩蒂在修剪\|**玫瑰**时不小心划破了手指. |
| Plausible and related | 粗心的佩蒂在修剪\|**鲜花**时不小心划破了手指. |
| Plausible and unrelated | 粗心的佩蒂在修剪\|**刘海**时不小心划破了手指. |
| Implausible and related | 粗心的佩蒂在修剪\|**爱情**时不小心划破了手指. |
| Implausible and unrelated | 粗心的佩蒂在修剪\|**海洋**时不小心划破了手指. |

Note:
In the boundary paradigm, bold words are previewed words, which are not bold in the experiment. The sentence translates as "*Careless Petty accidentally cut her finger while pruning **roses/flowers/bang/love/marine***". When readers fixations across the invisible boundary between the pretarget words and target words "|", the preview word 玫瑰 (rose), 鲜花 (flower), 刘海 (bang), 爱情 (love), 海洋 (marine) changes as the target word 玫瑰 (rose).

different word categories and comprised 65% nouns, 21% verbs, and 14% adjectives. Five previews were matched for each target word: identical to the target word (identical), plausible continuation of the preceding sentence context and semantically related to the target word (plausible and related), plausible and unrelated, implausible and related, and implausible and unrelated. None of the implausible previews violate syntactic rules because they are in the same grammatical class (word category) as the target words. Thus, the implausible previews constituted semantically implausible continuations of the preceding sentence context but remained syntactically acceptable. The experimental materials are shown in Table 1. There was no significant difference in word frequency ($F = 0.12$, $p = 0.95$) or visual complexity (as measured by the number of strokes, $F = 0.78$, $p = 0.53$) under the five preview conditions. Word frequency was based on the SUBTLEX-CH database (*Cai & Brysbaert, 2010*).

Each target word was embedded into a sentence with a length of 17–25 words (as shown in Table 1). There are 80 groups of sentence frames, each containing five preview conditions. None of the target words appeared in the first or last four words, and all the pretarget words (the word before the target word) were two-character words. By using the boundary paradigm, an invisible boundary was set between the pretarget word and the target word. A preview stimulus with one of five conditions was presented at the target word position before the reader's fixation crossed the boundary. Once the reader made a saccade across the boundary, the preview word was replaced by the target word.

The boundary paradigm (*Rayner, 1975*) is shown in Table 1. All sentences containing five preview conditions were counterbalanced across five blocks; each block contained sentence frames for all conditions, and each sentence frame was presented to each participant only once.

## Stimulus norming

The result of stimulus norming is shown in Table 2.

Due to the lack of grammatical units such as suffixes in the Chinese language, many words have flexible syntactic categories, and identifying the category is heavily dependent on context. Therefore, it is important to ensure that readers can infer the category of the upcoming word as much as possible from the preceding sentence; otherwise, they could

**Table 2 Properties of the experimental materials and norming data.**

| Variables | Preview conditions | | | | |
|---|---|---|---|---|---|
| | Identical | Plausible and related | Plausible and unrelated | Implausible and related | Implausible and unrelated |
| Stimulus properties | | | | | |
| Examples | 玫瑰 | 鲜花 | 刘海 | 爱情 | 海洋 |
| Strokes | 17.55 (4.58) | 17.00 (4.05) | 16.69 (3.50) | 16.74 (4.60) | 16.89 (3.57) |
| Frequency | 38.82 (70.23) | 39.05 (122.58) | 32.36 (107.75) | 33.00 (107.75) | 29.64 (58.04) |
| Norming data | | | | | |
| Predictability | 0.02 (0.04) | 0.01 (0.03) | 0.01 (0.03) | 0.00 (0.00) | 0.00 (0.00) |
| Semantic relatedness | —— | 6.12 (0.45) | 2.22 (0.33) | 6.08 (0.44) | 2.16 (0.44) |
| Plausibility | 6.07 (0.45) | 6.07 (0.50) | 6.08 (0.48) | 1.71 (0.33) | 1.71 (0.36) |

**Note:**
The standard deviation of the mean is shown in parentheses.

experience syntactic violations when the category of the word differs. To ensure that this manipulation is effective, two kinds of word category ratings were performed (*Yang et al., 2009*).

The first kind involved the predictability of the word category. Twenty participants predicted the word category of the next word according to the presented sentence frame up to the pretarget word. The results showed that 96% of the nouns were predicted to be nouns, 97% of the verbs were predicted to be verbs, and 94% of the adjectives were predicted to be adjectives. The second involved the word category decision. Participants had to decide the category of the preview word according to the given sentence framework up to (and including) the preview word. The rating results of another 20 participants showed that 98% of the nouns were confirmed to be nouns, 97% of the verbs were confirmed to be verbs, and 98% of the adjectives were confirmed to be adjectives. The two ratings ensured that our syntactic manipulation of the preview words was valid.

Twenty participants were recruited for the cloze task used to measure the predictability of the five preview conditions, in which the participants were asked to write the word that is most likely to come next according to the given sentence frame up to the pretarget word. The results showed that the predictability of all five conditions was very low (<2% on average).

Another group of 20 participants rated the semantic relatedness of the preview and target words on a seven-point scale, and the results showed that the plausibly related and implausibly related previews had significantly higher semantic relatedness than the plausibly unrelated and implausibly unrelated previews (all $ts > 49.00$, $ps < 0.001$). Moreover, the differences in the semantic relatedness of the two related previews or the two unrelated previews did not reach significance (all $ts < 1.27$, $ps > 0.20$).

Another 20 participants rated the contextual plausibility of the sentence frames up to each preview word on a seven-point scale. The results showed that the identical, plausible previews were all significantly more acceptable than the implausible previews (all $ts > 61.68$, $ps < 0.001$); the differences between the two plausible previews and the identical

previews were not significant (all $|t|$s < 0.20, $p$s > 0.84); and there was no significant difference between the two implausible previews ($t = -0.06$, $p = 0.95$).

None of the undergraduate participants who rated the materials participated in the eye-movement experiment, and each participant only participated in one of the rating studies.

## Apparatus

An SR Eyelink 1000 plus eye tracker with a sampling rate of 1,000 Hz was used for eye-movement recording. A Dell LCD monitor with a refresh rate of 144 Hz, and a 1020 × 1080 pixel resolution was used for monitoring. Each character was displayed in a 30-point font with 40 × 40 pixels on the screen in black on a white background. The participants' eyes were 60 cm away from the screen, and each character subtended approximately 0.96 degrees of visual angle.

## Procedure

After the participants understood the experimental procedure, three-point eye movement calibration of the right eye was conducted (with a calibration error of less than 0.3 degrees of visual angle). After successful calibration, each trial started with a fixation point appearing in the screen center, and then, a fixation box appeared on the first character at the beginning of the left side of the sentence. Once the participants stably fixated on the fixation box, the sentence was presented on the screen. The three-point calibration was repeated after every fourth trial and during the experiment when necessary. Each participant was randomly assigned to one of the five blocks. In addition to 80 experimental sentences, there were 60 filler sentences included in each block, and 25% of the sentences were followed by a simple "yes" or "no" judgment to ensure that the participants completed the experiment carefully. The first six sentences were practice sentences, and the remaining 140 sentences were presented randomly. The experiment lasted approximately 40 min.

# RESULTS

## Analysis

The comprehension accuracy rate of all participants was higher than 90%, indicating that the participants carefully read the sentences. Fixations that were shorter than 60 ms or longer than 600 ms were not included in the analysis (*Yan et al., 2009*; *Yang et al., 2012*). Trials with the following conditions were excluded: (1) failed tracking due to the participants coughing or other causes (0.3% of total trials); (2) fewer than five total fixations on the sentence (2.6% of total trials); (3) blinking at the boundary region or when fixating on the target word during the first-pass reading (3.1% of total trials), and trials in which the display change was completed earlier than or delayed by more than 10 ms into a fixation (4.8% of total trials); and (4) as previous studies using the same paradigm, where trials with regressions from the pretarget or target words were eliminated, as they may
**Table 3 Eye movement measures of target words under different preview conditions.**

| Measures | Identical | Plausible and related | Plausible and unrelated | Implausible and related | Implausible and unrelated |
|---|---|---|---|---|---|
| Single fixation duration | 237 (3) | 248 (4) | 247 (4) | 266 (4) | 281 (5) |
| First fixation duration | 239 (3) | 254 (4) | 254 (4) | 271 (4) | 281 (4) |
| Gaze duration | 263 (4) | 297 (6) | 314 (7) | 333 (7) | 355 (7) |
| Skipping rate | 0.23 (0.42) | 0.21 (0.40) | 0.20 (0.40) | 0.19 (0.39) | 0.17 (0.38) |

**Note:**
Fixation durations were measured in milliseconds. The standard errors is shown in parentheses.

reflect incomplete parafoveal preprocessing of the preview words or incomplete foveal processing of the target words (*Pan, Laubrock & Yan, 2016*; *Yan et al., 2012*) (19% of total trials). The remaining 70.2% of the data were used for analysis (4,473 trials).

The following four commonly used eye-movement measures for investigating the parafoveal preview benefit were analyzed (*Rayner, 1998*; *Schotter, Angele & Rayner, 2012*): *first fixation duration* (FFD), the duration of the first fixation on the target word during the first-pass reading; *single fixation duration* (SFD), the duration of the fixation when there is only one fixation on the target word during first-pass reading; *gaze duration* (GD), the total time of all first-pass fixations on the target word; and *skipping rate* (SR), the probability of the target word being skipped during the first-pass reading.

The data were analyzed using the *lme4* package (*Bates et al., 2015*) in the R environment (*R Core Team, 2018*). Linear mixed-effects models (LMM) were used to analyze the fixation duration measures, and the generalized LMM was used to analyze the skipping rate measures. The model takes each variable and their interaction as fixed factors and includes the cross random effects of the participants and items. If the maximum model of random effects does not successfully converge, then the model is gradually reduced by first removing the correlation of the item and then removing the slope of the item. If the model still does not converge successfully, then the correlation and slope of the subjects are removed until the model successfully converges.

We analyzed the target word region and tested the following planned contrasts: (a) *the typical preview effect*, the difference between the average of nonidentical and identical previews; (b) *the plausibility preview effect*, the difference between plausible and implausible previews; and (c) *the semantic relatedness effect*, the difference between semantically related and unrelated previews. The log-transformed analysis and untransformed analysis of fixation duration measures obtained the same significance pattern. Therefore, we reported the untransformed model results for transparency. For all the analysis results, a $t/z$ value of greater than 1.96 indicated a significant difference ($p < 0.05$).

## Result

The descriptive statistics of each measure of the target word are shown in Table 3, and the analysis results of the linear mixed model are shown in Table 4.

**Table 4 Results for the mixed-linear model analysis of target words measures.**

| Measures | Fixed effect | b | SE | T/z | p |
|---|---|---|---|---|---|
| SFD | Identity | 23.51 | 4.15 | 5.67 | <0.001 |
| | Plausibility | 28.97 | 5.43 | 5.33 | <0.001 |
| | Relatedness | 7.87 | 4.91 | 1.60 | 0.11 |
| | Plausibility × Relatedness | 16.85 | 9.72 | 1.73 | 0.09 |
| FFD | Identity | 26.38 | 3.87 | 6.82 | <0.001 |
| | Plausibility | 22.67 | 4.41 | 5.14 | <0.001 |
| | Relatedness | 5.58 | 3.95 | 1.41 | 0.16 |
| | Plausibility × Relatedness | 9.75 | 7.62 | 1.28 | 0.20 |
| GD | Identity | 62.96 | 6.61 | 9.53 | <0.001 |
| | Plausibility | 37.67 | 8.30 | 4.54 | <0.001 |
| | Relatedness | 18.94 | 7.44 | 2.54 | 0.01 |
| | Plausibility × Relatedness | 3.71 | 13.48 | 0.28 | 0.78 |
| SR | Identity | −0.19 | 0.1 | −1.89 | 0.06 |
| | Plausibility | −0.18 | 0.08 | −2.27 | 0.02 |
| | Relatedness | −0.08 | 0.08 | −1.04 | 0.30 |
| | Plausibility × Relatedness | −0.11 | 0.16 | −0.68 | 0.50 |

***The typical preview effect***

There was a significant preview effect in all fixation duration measures ($ts > 5.67$, $ps < 0.001$) and a marginally significant preview effect in the skipping rate measures ($z = -1.89$, $p = 0.06$).

A different pattern occurred in the plausibility preview effect and the semantic relatedness preview effect. A significant plausibility preview effect was observed in all fixation duration measures ($ts > 4.54$, $ps < 0.001$) and in the skipping rate measures ($z = -2.27$, $p = 0.02$). The fixation durations of the target words were shorter in the plausible preview condition than in the implausible preview condition, and the readers were more likely to skip the target words when the preview words were plausible in the first-pass reading. However, the semantic relatedness effect was found only in the gaze durations ($t = 2.54$, $p = 0.01$), and no significant differences were observed between the semantically related and unrelated preview conditions in the other fixation duration measures ($ts < 1.60$, $ps > 0.11$) or the skipping rate measure ($z = -1.04$, $p = 0.30$). In addition, the interaction effect of preview plausibility and semantic relatedness did not approach significance across any of the fixation duration measures ($ts < 1.73$, $ps > 0.09$) or in the skipping rate measure ($z = -0.68$, $p = 0.50$).

## Supplementary analyses

### Bayesian analysis

According to the descriptive statistics in Table 3 and the inferential statistics in Table 4, the results show that there is not enough evidence to support the interaction effect between preview plausibility and semantic relatedness, even if there was a tendency toward an

interaction effect in the single fixation duration measure ($t = 1.73$, $p = 0.09$). To further test the reliability of this result, the LmBF () function in the BayesFactor package (0.9.12–4.2; *Morey & Rouder, 2018*) was used to conduct a Bayesian analysis of the fixation duration measures.

Since we focus on the interaction effect between preview plausibility and semantic relatedness, we calculate the ratio of the Bayesian factor (*BF*) of the full model (including the main effect and the interaction effect between the two variables) to the model containing only two main effects. If the *BF* is less than 1, the null hypothesis is supported, indicating that there is no interaction effect. If the *BF* is greater than 1, the alternative hypothesis is supported, indicating that there is an interaction effect. If the *BF* is greater than 3, it provides substantial evidence to support the full model and indicates that there is an interaction effect (*Wagenmakers et al., 2018*). The results suggest that there is insufficient evidence to support the interaction effect between preview plausibility and semantic relatedness (SFD: *BF* = 0.25; FFD: *BF* = 0.25; GD: *BF* = 0.26).

## DISCUSSION

The present study investigated the underlying mechanisms of SPB in Chinese reading by precisely manipulating the preview plausibility and semantic relatedness and controlling for the syntactic plausibility variable. Our study found that there was a robust plausibility preview effect in the first-pass reading measures, suggesting that readers can benefit from parafoveal semantically plausible previews compared to implausible previews in reading. However, the significant effect of the semantic relatedness was found only in the gaze duration measure, and the Bayesian analysis showed that there was no significant interaction effect between preview plausibility and semantic relatedness. Our results show that preview plausibility may exert a stronger and earlier effect on SPB than semantic relatedness, suggesting that preview plausibility preferentially influences SPB in Chinese reading. This pattern of results provides direct evidence for the context fit account.

The plausibility preview effect was observed in all measures used in the present study, suggesting that the SPB may result from the plausibility of contextual fitting between the preview word and the preceding context. The pattern of results is consistent with the findings of a previous Chinese reading study (*Yang et al., 2012*) and those of alphabetic script reading studies (*Veldre & Andrews, 2016*), in that plausible previews that fit the context of the sentence can facilitate the processing of target words, while implausible ones interfere with that processing. More importantly, our study also found a robust plausibility preview effect when the syntactic plausibility variable was strictly controlled, indicating that the plausibility preview effect could be observed without syntactic violations during Chinese reading. This observation is compatible with the findings for alphabetic languages (*Veldre & Andrews, 2018*), which show that readers benefit from previews that are both syntactically and semantically plausible relative to syntactically acceptable but semantically implausible previews during first-pass reading. Therefore, the present study further clarifies that semantic preview plausibility plays a key role in the SPB effect.

We also observed the semantic relatedness effect in gaze duration, suggesting that semantic relatedness may also play a role in the occurrence of SPB. This result differs from

those of previous studies (*Li, Sun & Wang, 2022*; *Yang et al., 2012*). By orthogonally manipulating the two variables and strictly controlling the syntactic plausibility of that all preview words, we found a significant semantic relatedness effect that is independent of preview plausibility. However, despite the semantic relatedness effect had an effect on SPB, the preview benefit was much greater for the plausibility variable ($t = 4.54$) than for the relatedness variable ($t = 2.54$) in the gaze duration, as shown in Table 4. Thus, we may say that semantic relatedness plays a weaker role in the SPB. Moreover, gaze duration is a measure that is conducted slightly later than the other measures, such as the single and first fixation duration measures. Therefore, we may also say that semantic relatedness plays a weaker role at the later stages of the SPB.

No reliable interactive effects were observed between the two independent variables in the present study, suggesting that the occurrence of the SPB seems to be independently affected by the preview plausibility and semantic relatedness. This pattern of results may be due to preview plausibility and semantic relatedness playing different roles in sentence processing. The former involves the integrative comprehension of preactivated semantic information and the preceding sentence context (*Veldre & Andrews, 2016*), while the latter involves the identification of the preview and the target words. For example, the main effect of preview plausibility observed in this study indicates that an acceptable/plausible continuation preview word that fits with the sentence context leads to shorter first-pass reading times of the target word, whereas an implausible preview word that is inconsistent with the semantic representation of the sentence being developed leads to longer reading times. This dynamic plausibility integration process occurs independently of the semantic relationship between the preview and target words. After all, the reader has already initiated a saccade plan (reflected in the skipping rate measures) based on the plausibility of the preview word before fixating on the target word. Therefore, it can be concluded that preview plausibility has an impact on the SPB independent of semantic relatedness, which is consistent with the research results of parafoveal semantic processing in English reading (*Antúnez et al., 2022*; *Veldre & Andrews, 2016*).

Combining the finding obtained in the present study detailing that preview plausibility has an earlier and stronger effect on SPB than semantic relatedness, we prefer to use the contextual fit account to explain the SPB effect. Unlike alphabetic scripts, there are no obvious physical markers (such as intercharacter/interword spaces) between characters/words for word segmentation in written Chinese text, so readers need to rely heavily on the sentence context to perform word segmentation (*Chen, 1992*; *Yang et al., 2012*). This high dependence on sentence context requires readers to process and integrate the previewed information with the sentence context on-line (*Yang et al., 2012*), and roughly evaluate whether the preview word is an acceptable continuation of the sentence context. Therefore, preview plausibility affects the SPB at a relatively early stage. Moreover, it has been suggested that readers activate a set of alternative continuation words according to the sentence context and prior knowledge or experience during reading (*Levy, 2008*; *Luke & Christianson, 2016*). In this case, both the preview and target words are plausible continuations of the sentence context and are in the alternative set (*Veldre et al., 2020*). Therefore, context fitting allows for previewing a plausible word to be compatible with the

target word, which will not result in difficulties in lexical processing and integrated comprehension and facilitates the processing of the target word (*Veldre & Andrews, 2018*; *Veldre et al., 2020*). However, implausible previews always violate the reader's expectation of the semantic representation that is incrementally constructed from the sentence context, which rapidly leads to difficulties in word integration and sentence comprehension, resulting in longer fixation times on the target word (*Veldre & Andrews, 2018*).

On the other hand, the effect of semantic relatedness on the SPB mainly relies on the semantic spreading activation between the preview and the target words rather than on the sentence contextual information or knowledge experience. Our research shows that the effect of semantic spreading lags behind that of contextual information in the process of word identification for sentence comprehension. Specifically, semantic spreading activation begins after the reader processes the semantic information of the preview word (*Li, Sun & Wang, 2022*). In the boundary paradigm, the effect of the preview word on target word identification occurs after the reader's eyes cross the invisible boundary and fixate on the target word. However, readers may preactivate a set of possible upcoming words based on the prior sentence context information and knowledge experience before fixating on the target word (*Veldre et al., 2020*). This preactivation based on sentence context affects the processing of target words before that based on the semantic relatedness of the preview words. Therefore, we conclude that preview plausibility exerts a more preferential effect on the SPB than semantic relatedness in the process of Chinese reading.

These findings might have some implications for developing reading models. Based on previous research (*Li, Sun & Wang, 2022*; *Zhu, Zhuang & Ma, 2021*; *Yang et al., 2012*), we discovered that semantic plausibility plays a critical role in sentence comprehension by independently manipulating preview plausibility and semantic relatedness and excluding the interference of syntactical implausibility on SPB. Our results suggest that preview plausibility plays a significant role during sentence comprehension. The postlexical integration stage (*I*) introduced by E–Z Reader 10 (*Reichle, Warren & McConnell, 2009*) provides a possible mechanism for the plausibility preview effect. Briefly, the model assumes that the integration process follows the completion of lexical processing for each word to roughly evaluate whether the word is compatible with the incrementally constructed sentence semantic representation (*Reichle et al., 2013*). A delay in or failure of this integration results in the cancellation of the forward saccade plan. Combined with our findings, this suggests that readers can obtain semantic information from the parafoveal word and rapidly evaluate whether the previewed word is a plausible continuation of the developing sentence context. This may be due to sentence contextual information and knowledge experience quickly contributing to the identification of the upcoming word and affecting the integrated comprehension of the word and the sentence context. On the other hand, semantic spreading activation begins only after the reader obtains the semantic information of the previewed word, which occurs later than the effect of the developing sentence context information on word identification. Therefore, preview plausibility takes priority over semantic relatedness and exerts a stronger effect on lexical processing. Our findings provide empirical support for the eye-movement control model to explain the parafoveal SPB effect and contribute to the development of eye-movement control

models in reading. Future research can also more directly investigate the influence of parafoveal syntactic information on SPB and develop a model that can systematically explain the parafoveal processing mechanism of high-level information in the reading.

## CONCLUSIONS

In conclusion, this study aimed to gain a better understanding of the SPB. Our study found that preview plausibility plays an earlier and more comprehensive role in the occurrence of the SPB, while semantic relatedness has a much later and limited effect on it. The results reveal that the plausibility relationship between the preview words and their contextual sentence may preferentially affect the SPB, supporting the context fit account.

### Funding

This research was supported by the National Natural Science Foundation of China (32271119). The funders had no role in study design, data collection and analysis, decision to publish, or preparation of the manuscript.

### Grant Disclosures

The following grant information was disclosed by the authors:
National Natural Science Foundation of China: 32271119.

### Competing Interests

The authors declare that they have no competing interests.

### Author Contributions

- Lijuan Zhang conceived and designed the experiments, performed the experiments, analyzed the data, prepared figures and/or tables, authored or reviewed drafts of the article, and approved the final draft.
- Zhiwei Liu conceived and designed the experiments, authored or reviewed drafts of the article, and approved the final draft.
- Sainan Zhao conceived and designed the experiments, authored or reviewed drafts of the article, and approved the final draft.
- Jingxin Wang conceived and designed the experiments, authored or reviewed drafts of the article, and approved the final draft.

### Human Ethics

The following information was supplied relating to ethical approvals (*i.e.*, approving body and any reference numbers):

This study was approved by the Research Ethics Committee of the Faculty of Psychology, Tianjin Normal University.

## Data Availability

The raw data is available at figshare: Zhang, LIjuan (2022): target_IA_2*2.csv. figshare. Dataset. https://doi.org/10.6084/m9.figshare.21346200.v4.

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
