# Peer review of "Semantic plausibility preferentially affects the semantic preview benefit in Chinese reading: evidence from an eye-movement study"

_PeerJ, doi:10.7717/peerj.15291_

## Round 0.1 · original submission · Minor Revisions

· Academic Editor

Minor Revisions

I have received reviews from two experts in the field. Reviewer 1 is Erik Reichle. Both reviewers are positive about the manuscript but have identified rather minor issues that should be addressed before the piece meets threshold for publication. The most major critique from both is that the piece would benefit from a clearer articulation of the rationale for carrying out your experiment.

I request that you add a statement to the paper confirming whether, for all experiments, you have reported all measures, conditions, data exclusions, and how you determined your sample sizes. You should, of course, add any additional text to ensure the statement is accurate. This is the standard reviewer disclosure request endorsed by the Center for Open Science [see http://osf.io/project/hadz3]. I include it in every review.

I identified some typos and grammar problems beyond those identified by the reviewers that should be remedied in your revision:
- line 67: I think "processing of target word than" should be "processing of the target word more than"
- line 106: "knowledge field gaps" should just be "knowledge gaps"
- line 107: "which may has contributed" should be "which may have contributed"
- line 130: "latter" should be "later"
- line 171: remove the first instance of "target"
- line 233: You include the screen resolution and refresh rate as if it corresponds to the tracker. Move this information so it is paired with the monitor details.
- When reporting p-values, p is frequently capitalized when it should not be

I look forward to reading a revision of this work and anticipate that I will be able to make a quick decision after these minor issues have been remedied.

·

Basic reporting

The basic reporting was overall quite good. I have several minor suggestions to help make the manuscript stronger, organized by line(s):
57: Why Chinese? Out of convenience, because the authors are in China? Or because some property of written Chinese allows the research question to be addressed in a manner not possible in English?
65: "from parafovea" --> "from the parafovea"
72-73: The sentence beginning with "According to..." is not quite correct; a researcher or theory might hold or be consistent with an opinion, but not a theoretical construct (i.e., semantic spreading activation).
106-108: This sentence is really useful because it frames the purpose of the study. Consider moving it forward to someplace earlier in the manuscript.
154: "semantic spreading? --> "semantic activation spreading"
221: "The other" --> "Another"
255: "if display" --> "if the display"
256: "use" --> "using"
257: "as it may" --> "as they may"
268: "-Linear" --> "Linear"
289: "Different stories" --> "A different pattern"
304: Table 1 should be Table 3.
305: Table 2 should be Table 4.
315-317: The criteria used for evaluating BFs is unclear.

Experimental design

The experimental design, implementation, and execution all seem fine. The work was conducted to a high technical and ethic standard. The method was also described well enough to replicate. One aspect of the experiment was a little odd, however, and should probably be addressed with an additional sentence or two. The experiment was done explicitly to control for the possible syntax violations that limit the interpretation of previous experiments, as described in lines 106-108. It is therefore somewhat odd that the experiment was conducted in Chinese--a language in which syntax plays a minor role compared to semantics. Some acknowledge of this fact seems appropriate.

Validity of the findings

The results of the experiment will contribute to our understand of semantic preview and the role played by plausibility in generating semantic preview effects. As far as I can tell, the data are sound, analyzed correctly, etc. The conclusions also seem fair, and are interesting but not overstated.

Additional comments

Overall, I think this manuscript is well written and that the experiment described therein is worthy of publication.

Reviewer 2 ·

Basic reporting

The manuscript is well-written and showcases clear and professional use of language throughout. In accordance with PeerJ's Data Sharing policy, it is strongly recommended that the authors make their raw data and codes available for sharing.

Experimental design

no comment.

Validity of the findings

no comment.

Additional comments

Major concern:
The authors need to provide a more robust justification for their study. Currently, two eye-movement studies have explored the impact of preview plausibility and semantic relatedness in Chinese reading through the boundary change paradigm (Li et al., 2022; Yang et al., 2012). Yang et al., (2012) manipulated semantic relatedness in Experiment 1 and both preview plausibility and semantic relatedness in Experiment 2. The study found no effect of semantic preview in Experiment 1 and a significant impact of plausibility, as well as some evidence for a semantic preview benefit, in Experiment 2. Li et al., (2022) used highly constraining sentences and followed a similar experimental design as Yang et al., (2012) and concluded that semantic relatedness only had an effect when the preview words were plausible.
The current paper's contribution to the literature is unclear given these previous studies. The authors assert that syntactic plausibility was controlled, but also acknowledge that the study does not examine the impact of parafoveal syntactic information on SPB. Therefore, the authors should provide a stronger rationale for their study and explain how it fills a gap in the literature. A discussion of how the results add to previous studies is also necessary.

Minor comments:
- Line 24. semantic preview benefit, the p is missing.
- Line 164. + Line 170. Provide justification for sample size.
- Line 254. Please specify if this was only during first-pass reading or if it also included blinks on the target word during rereading.
- Lines 311 to 321. In this passage, I think you mean “additive” not “addictive”.
- Ensure that all letters used to indicate statistical symbols are italicized.

Annotated reviews are not available for download in order to protect the identity of reviewers who chose to remain anonymous.

---

## Round 0.2 · accepted · Accept

· Academic Editor

Accept

Thank you for submitting your revisions. Given that the reviewers on the first round were quite positive about the manuscript, I elected not to send it out for review again. I have read the revised manuscript and your responses to the reviewers. I am satisfied that you have adequately addressed all of the reviewers' concerns, and I am happy to accept this manuscript for publication at PeerJ.